# 3D Printed Chitosan/Alginate Hydrogels for the Controlled Release of Silver Sulfadiazine in Wound Healing Applications: Design, Characterization and Antimicrobial Activity

**DOI:** 10.3390/mi14010137

**Published:** 2023-01-04

**Authors:** Carlo Bergonzi, Annalisa Bianchera, Giulia Remaggi, Maria Cristina Ossiprandi, Ruggero Bettini, Lisa Elviri

**Affiliations:** 1Food and Drug Department, University of Parma, Parco Area delle Scienze 27/a, 43124 Parma, Italy; 2Department of Veterinary Science, University of Parma, Strada del Taglio 10, 43126 Parma, Italy

**Keywords:** chitosan, alginate, silver sulfadiazine, 3D printed hydrogels, antimicrobial activity, wound healing

## Abstract

The growing demand for personalized medicine requires innovation in drug manufacturing to combine versatility with automation. Here, three-dimensional (3D) printing was explored for the production of chitosan (CH)/alginate (ALG)-based hydrogels intended as active dressings for wound healing. ALG hydrogels were loaded with 0.75% w/v silver sulfadiazine (SSD), selected as a drug model commonly used for the therapeutic treatment of infected burn wounds, and four different 3D CH/ALG architectures were designed to modulate the release of this active compound. CH/ALG constructs were characterized by their water content, elasticity and porosity. ALG hydrogels (Young’s modulus 0.582 ± 0.019 Mpa) were statistically different in terms of elasticity compared to CH (Young’s modulus 0.365 ± 0.015 Mpa) but very similar in terms of swelling properties (water content in ALG: 93.18 ± 0.88% and in CH: 92.76 ± 1.17%). In vitro SSD release tests were performed by using vertical diffusion Franz cells, and statistically significant different behaviors in terms of the amount and kinetics of drugs released were observed as a function of the construct. Moreover, strong antimicrobial potency (100% of growth inhibition) against *Staphylococcus aureus* and *Pseudomonas aeruginosa* was demonstrated depending on the type of construct, offering a proof of concept that 3D printing techniques could be efficiently applied to the production of hydrogels for controlled drug delivery.

## 1. Introduction

Personalized medicine and tailored therapies are in growing demand in drug manufacturing as they could offer higher drug efficacy as well as higher safety [1,2]. To meet the requirements of personalized medicine, drug formulation must rely on manufacturing processes that combine the need for customization with industrial requirements, guaranteeing law-compliant standards of accuracy, flexibility and automation [3]. Briefly, 3D printing is one of the most promising approaches to overcoming these issues. Many 3D printing techniques have been applied to the production of solid dosage forms, mainly intended for oral drug delivery [4,5]. Printing processes commonly applied to pharmaceutical products are stereolithography (SLA) and fused-deposition modeling (FDM): in the first technique, a light source causes the polymerization of photocrosslinkable polymers [6], whereas the second one relies on the melting of polymers that are then deposited on a build plate layer-by-layer [1]. In recent years, many GRAS-approved materials have been successfully tested using FDM [1,7]. Moreover, these techniques allow the production of solid dosage forms with rigid or flexible structures, which are desirable candidates for wound dressings [8] or tissue engineering applications. In a recent review by Jang et al. [9], advances in the 3D printing of hydrogels are described, including low-temperature deposition associated with freeze-drying.

In this paper, an alternative to this technique, based on cryogenic prototyping followed by ionotropic gelation, previously described by Elviri et al. [10], was used to produce 3D printed polymeric drug delivery systems [11] in the form of hydrophilic gels. Two natural biopolymers, alginate (ALG) and chitosan (CH), were chosen, having useful characteristics such as safety, biodegradability, and biocompatibility [12,13,14]. Alginate is an anionic polysaccharide consisting of linear copolymers of β-1,4 linked D-mannuronic acid and β-1,4 linked L-guluronic acid units. It is commonly produced by brown algae, and it is extracted from marine biomass mainly by alkaline extraction. The copolymeric nature of alginate provides high viscosity, gelling properties at mild temperatures and high stability, physical characteristics that paved the way for the employment of this polysaccharide in different applications, including tissue engineering and drug delivery [12,13,14]. ALG took advantage of the fast ionic crosslinking behavior in the presence of bivalent cations (e.g., Ca^2+^) resulting in the shape maintenance of the hydrogel constructs.

In the pool of naturally occurring polysaccharides, chitosan plays a central role in 3D printing for wound healing applications. Chitosan is a natural linear polysaccharide obtained from the deacetylation of chitin extracted from living organisms, such as shrimps and fungi. CH is biocompatible and biodegradable, and its degradation products are not toxic. 

Their chemical similarity with glycosaminoglycans composing extracellular matrices makes them suitable as excipients in drug delivery systems and medical devices [15], including commercially available advanced wound dressings [16]. The combination of benefits of both natural materials allowed to obtain biocompatible materials with tunable mechanical properties and controlled drug release properties. 

Ideally, dressings for wound treatment should be changed as infrequently as possible and a sustained release of antimicrobial agents is a desirable feature to control bacterial proliferation as well as the capacity of the medication to be integrated by the tissue [17]. Here, silver sulfadiazine (SSD), an FDA-approved active principle widely used for the therapeutic treatment of infected burn wounds, was selected as a drug model to be included in the hydrophilic scaffolds. SSD exhibits fast and broad-spectrum antimicrobial activity against both Gram-positive and Gram-negative bacteria [18,19]. It acts as a competitive inhibitor of PABA, which causes an interruption of folic acid metabolism, and therefore of the synthesis of bacterial DNA, whereas silver ions act on the bacterium’s energy system [20]. Moreover, SSD has been demonstrated to be effective also against multidrug-resistant organisms [21,22]. The development of active dressings able to control SSD release over time is of great interest for applications in wound healing, as demonstrated by the several publications reported on this topic [23,24,25,26,27,28,29,30,31,32,33,34]. By considering about 70 published papers on the use of SSD in wound healing, an interesting review demonstrated, through a random effects model, that healing goes faster (−2.72 days; 95% confidence interval: −4.99, −0.45) in SSD-treated groups when compared to the control group over 21 days [29].

In such a context, formulations including SSD in combination with natural polysaccharides for the preparation of hydrogels in wound healing applications are under continuous investigations for their high biocompatibility, flexibility and efficacy. As examples, recently, chitosan–SSD nanoparticles were efficiently included in PVP-K30 support to obtain controlled drug release nanofibers active against both Gram-negative (*Pseudomonas aeruginosa, Escherichia coli, Acinetobacter baumannii*) and Gram-positive bacteria (*Staphylococcus aureus* and *Enterococcus faecalis*) [23]. In another study, alginate was used in combination with chitosan to formulate nanogels containing SSD by using a factorial experimental design to optimize the drug release and the final dressing showing higher therapeutic efficacy in vivo when compared to market products [34].

The application of 3D printing to the production of hydrogels based on chitosan and/or alginate reported in the literature are mainly conceived for tissue engineering [35] whereas few reports describe their use as drug delivery systems of SSD, which all rely on classical casting methods. 

Boateng et al. [36] described the development of freeze-dried alginate/gelatin bio-polymeric wafers for potential application on infected chronic wounds capable of releasing SSD within 7 h at concentrations higher than the MIC but not in a controlled manner. Shao et al. [37] proposed genipin crosslinked chitosan sponges loaded with SSD with excellent antibacterial performances, whereas Fajardo et al. [38] loaded SSD into chitosan/chondroitin sulfate films, reaching the maximum drug release in about 12 h. 

In this paper, in order to add innovative insights into this issue, CH/ALG scaffolds loaded with SSD with four different 3D printed architectures were designed and characterized from a physicochemical point of view. The kinetic release profile of SSD was evaluated as a function of the 3D printed hydrogel geometries. Finally, the effectiveness in terms of antimicrobial activity in vitro against *Staphylococcus aureus (SA)* and *Pseudomonas aeruginosa (PA)* was assayed.

## 2. Materials and Methods

### 2.1. Reagents

Chitosan (ChitoClear^®^ Fg90 TM1874–CAS 9012-76-4, degree of deacetylation 95%; molecular weight by gel permeation chromatography 150–200 kDa; allergen free, water-insoluble, soluble in acid media) was from Primex (Primex EHF, Siglufjordur, Island). Sodium alginate (Ph.Eur. grade; molecular weight by gel filtration chromatography (GFC) 180–300 kDa; slowly soluble in water) was from Carlo Erba (Carlo Erba Reagents Srl, Milan, Italy). Trypsin, acetonitrile, phosphoric acid, potassium hydroxide, sodium chloride, calcium chloride, ammonia, tris methylamine and raffinose pentahydrate were from Sigma-Aldrich (St. Louis, MI, USA). Dulbecco’s minimal essential medium (DMEM) was from Lonza. Penicillin, streptomycin, and fetal bovine serum were from Euroclone (Euroclone Spa, MI, Italy).

Ultrapure water was obtained by using Flex ultra-pure water apparatus (Elga Veolia Water Technologies, Milan, Italy).

### 2.2. Preparation of 3D Printed Scaffolds

Chitosan powder was dispersed in acetic acid aqueous solution (2% *v/v*) at the concentration of 6% *w/v*. The suspension was stirred magnetically for 24 h; then, after the complete dissolution of the polymer, raffinose pentahydrate was added as a rheological agent at the final concentration of 290 mM and stirred for a further 12 h. Sodium alginate was suspended at a concentration of 6% (*w/v*) in a mixture of ultrapure water and 28% ammonia (95:5, *v/v*) to help the polymer dissolution.

Three-dimensional objects were designed and produced by means of a 3D printer built in-house that exploits freeze deposition to give shape to polymeric solutions, as previously described [39]. Briefly, a grid (1.5 × 1.5 cm) of parallel overlapped filaments, with a distance of 200 μm, was drawn by Solidworks^TM^ (Dassault Systems, Waltham, MA, USA), and the file was converted by Slic3r^TM^ (https://slic3r.org, accessed on 20 December 2022, Italy) into a machine code (.gCode) readable by the 3D printer. The 3D printer works by extruding the material loaded onto a 5 mL syringe through a 26 G needle (inner diameter 192 μm) layer by layer. The thickness of the hydrogels was customized, and 5 or 20 layers were printed, resulting in 1 mm- or 4 mm-height scaffolds, respectively.

At the end of each printing process, the frozen polymeric hydrogel underwent ionotropic gelation to maintain its structure. In the case of chitosan hydrogels, gelation occurred by exposure to ammonia vapors [38], whereas sodium alginate hydrogels were gelled by direct immersion in a CaCl_2_ aqueous solution (3%, *w/v*). After gelling, the hydrogels were washed twice in 80 mL of ultrapure water for 10 min until the pH of the aqueous solution was neutral, indicating the removal of the gelling agent.

SSD stability was preliminary checked in the presence of polymers and their co-solvents in solution before the printing of hydrogels, revealing drug degradation: for this reason, the drug was loaded onto printed calcium alginate hydrogels by directly soaking them in a 5% ammonia aqueous solution, into which, SSD was dissolved at a concentration of 0.75 % (*w/v*). The choice of using ammonia for the preparation of the loading solution was due to the very low SSD solubility in water and other organic solvents. Each hydrogel was soaked in 3 mL of drug solution for 3 h in amber containers on an orbital shaker (IKA, Shuttler MTS 4). After soaking, the hydrogels were stored in amber vials at 4 °C until use.

### 2.3. Water Content and Elasticity

As the water content and hydration state of hydrogels are critical for drug loading and release, the total water content of each kind of hydrogel was measured. Five replicates of alginate and chitosan hydrogels (five-layer scaffolds were used as a reference system) were produced as described above, washed once with ultrapure water, then gently blotted on filter paper to remove the excess water present within the pores and finally accurately weighed. Then, the hydrogels were totally dehydrated in an oven at 40 °C and 200 mbar of pressure (Gallenkam vacuum oven; Gallenkamp & Co. Ltd., London, UK). Once anhydrous, hydrogels were re-weighed, and their water content was calculated using the following formula [40]:*% of water = 100 − (W_a_ × 100)/W_w_*
where:*W_a_*: weight anhydrous*W_w_*: weight wet.

A traction dynamometer (Aquati AG MC1, Aquati Srl, Arese, Milan, Italy) was used to perform traction tests on alginate and chitosan hydrogels (size 48 × 14 mm, 20 layers thick). The traction speed was set at 25 mm/min using a five daN top head cell and setting a distance between clips of 25 mm. The applied force and movement were digitalized by PowerLab 4/35 software (released 24 October 2013) and registered by LabChart^®^Pro software v. 8.0 (ADInstruments, Austin, TX, USA). The stress applied and the elongation of the specimen were registered constantly. The thickness of the specimen was measured by a digital thickness feeler (Mitutoyo Corporation, Takatsu-ku, Kawasaki, Kanagawa) before traction tests. Elongation at break (% strain) and Young’s modulus were calculated from the relevant stress–strain curves, taking into consideration the linear portion, corresponding to the elastic behavior of the specimens. In particular, Young’s modulus was calculated using the formula:E = σ/ε
where σ corresponds to stress (applied force/cross-section area) and ε to strain (net elastic elongation) [40].

### 2.4. Scanning Electron Microscopy (SEM) Analysis

For a deep morphological characterization, SEM analyses were conducted. Five-layer chitosan and alginate hydrogels were prepared; in particular, alginate samples with and without SSD were compared. The hydrogels were dehydrated by immersion twice for 10 min in increasing grades of ethanol from 50° to absolute. The hydrogels were then dried by the critical point drying technique (Balserz Union, Lake Butler, FL, USA) to obtain anhydrous samples avoiding structure deformations. The anhydrous hydrogels were then fixed on support using double-sided carbon tape, sputter coated (E5100, Polaron, Quorum Technologies Ltd., Leves, UK) with gold (thickness 60 nm) and observed by SEM (Philips 501, Philips, Eindhoven, The Netherlands). Digital photographs of the hydrogels were analyzed by means of ImageJ software v. 1.53 (National Institute of Health, NIH, Bethesda, MD, USA) randomly measuring the Feret’s diameter of 150 pores and determining their size distribution.

### 2.5. ATR FTIR Spectroscopy

Infrared ATR FTIR spectroscopy was performed in order to evaluate the presence of the drug and to characterize its interactions with calcium alginate. Spectra were taken by a Nicolet 5700 (Thermo Scientific, Waltham, MA, USA) in the range of 400–4000 wavenumbers (cm^−1^) with a resolution of 30 scans per second. The spectra of the SSD anhydrous powder, dried 3D alginate hydrogels loaded with SSD and dried 3D alginate hydrogel without the drug were collected.

### 2.6. Hydrogel Combinations to Modify Drug Release

Different hydrogel combinations were designed and realized with the aim of investigating their influence on the release kinetics and amount of SSD. The drug was loaded in five-layer alginate hydrogels, which were then assembled with chitosan hydrogels by using alginate drops to attach the upper and lower parts of the different scaffolds. The CH/ALG constructs tested were designed as follows: (A) 3D printed five-layer alginate loaded with SSD; (AF) 3D printed five-layer alginate + chitosan film; (5CA) 3D printed five-layer chitosan hydrogel + 3D printed five-layer alginate; (20CA) 3D printed twenty-layer chitosan hydrogel + 3D printed five-layer alginate (20CA) (Figure 1). The film (layer thickness of 3 mm) in the construct (AF) was prepared by pouring chitosan solution into a mold, freezing it at −20 °C and then immersing it in a 1.5 M KOH solution for gelation [41].

### 2.7. In Vitro Release Tests

Release tests were performed in triplicate using vertical diffusion Franz cells with a diffusion area of 3 cm^2^ assembled to simulate the behavior of an exudating wound. The donor and receiver parts were separated by a regenerated cellulose membrane as a physical barrier to support the hydrogels (diameter of pores 0.45 μm), previously boiled in ultrapure water for 1 h to remove the air within the micropores and contingent contaminants. The receiver chamber was filled with 20 mL of a degassed buffer whose composition was conceived as simulated wound fluid (SWF) [42] composed of CaCl_2_ anhydrous 0.02 M, NaCl 0.4 M and tris methylamine 0.08 M (final pH 7.5). The cells were then placed in a thermostat bath at 37 °C under magnetic stirring.

ALG and CH/ALG constructs were placed in the donor chamber of the Franz cell, covering the whole diffusion area with the chitosan side (if present) facing down, in contact with the membrane; no further buffer was added to the donor chamber, which was closed with a waterproof cover to prevent solvent evaporation. At determined time points (5, 10, 30, 60, 180 and 300 min), 400 μL samples were taken from the receiving cell and collected in amber vials; the withdrawn volume was restored by the same amount of fresh SWF. 

The amount of released drug was evaluated by performing liquid chromatography–spectrophotometric ultraviolet/visible (LC–UV/VIS) analysis. An LC Agilent 1200 binary pump (Agilent Technologies, Santa Clara, CA, USA was used. Chromatographic separation was achieved on a C18 (250 × 4.6 mm, 10 μm) column (Supelco, Sigma-Aldrich, St. Louis, MO, USA). The mobile phase consisted of water, acetonitrile and phosphoric acid (90:9.1:0.1%, *v/v*), the flow rate of the mobile phase was maintained at 1.0 mL/min and the detector wavelength at 254 nm. The injection volume was 20 μL. Each sample was analyzed in triplicate. For the quantitative analysis, a calibration curve in the 1 to 20 μg/mL concentration range was prepared in SWF [Y = 36.42(±0.04)X; r^2^ = 0.999]. 

### 2.8. Cytotoxicity Test

The effect of SSD release from the different assemblies was evaluated in terms of cytotoxicity on human fibroblast cells, isolated using the technique of explantation from a cutaneous biopsy. Human fibroblasts were grown in Dulbecco’s minimal essential medium with penicillin and streptomycin and completed with fetal bovine serum at 10%, at 37 °C and 5% CO_2_. For the assay, cells were detached by trypsinization and seeded into 48-well culture plates (Corning) in complete growth medium at a density of 75,000 cells per well.

Assemblies A, AF, 5CA and 20CA were prepared following the methods previously described [10], punched with a 6 mm-diameter circular stamp and subsequently sterilized by immersion in ethanol 70% (*v/v*) overnight. Each assembly was washed in a PBS solution to bring them to a physiological pH and to eliminate ethanol residues. A scaffold per well was put into contact with cell monolayers. After 24 h of incubation, the cytotoxicity was quantified using the metabolic dye resazurin. Briefly, the culture medium and scaffolds were removed, and the cells were rinsed with warm PBS before adding into each well 500 μL of a sterile solution of resazurin in PBS at a concentration of 10 mg/mL. After 3 h of incubation, 100 μL of the solution was sampled from each well and transferred into a 96-well black plate for fluorescence measurement. The samples were excited at 560 nm, and emission at 590 nm was recorded and compared to untreated control samples. The experiments were performed in triplicate. 

### 2.9. Antimicrobial Activity

Antimicrobial activity evaluation against two strains of bacteria (Gram^+^ and Gram^−^) was assessed. Multi-drug resistant strains of *Staphylococcus aureus* (ATCC 25923) and *Pseudomonas aeruginosa* (ATCC 27853) were considered for such tests as they are responsible for frequent infections in chronic wounds [43]. The Kirby–Bauer technique was adopted and ad hoc-modified to assay 3D printed hydrogel scaffolds as a drug-loaded core of biocidal medicated dressings [44]. Briefly, 6 mm diameter stamps were used to obtain circular drug-loaded alginate scaffolds (the same size as the antibiogram discs) and kept in the fridge at 4 °C prior to the experiment’s start. At the same time, bacteria were seeded in pure culture and inoculated in Mueller Hinton Broth at 37 °C in aerobic conditions for 1–2 h (0.5 McFarland). Bacterial suspension was seeded using sterile tampons on a Mueller Hinton Agar terrain (carefully covering the entire Petri dish), and then scaffolds were applied using sterile forceps. Each test was conducted in duplicate. A negative and positive control were established: a negative one with the scaffold only, and a positive control consisting of a Petri dish seeded with bacteria but free of scaffolds. Finally, all the plates were incubated at 37 °C for 18–24 h. The result evaluation was based on the presence/absence of the inhibition ring, followed by its diameter measurement. The bacterial sensibility to the antimicrobial specimen was directly proportional to this latter parameter.

### 2.10. Statistical Analysis

Statistical analysis was carried out using Microsoft Excel software v. 16.68 (Microsoft Corporation, WA, USA). Data are given as mean ± standard deviation (SD). A value of *p* < 0.05 was considered statistically significant.

## 3. Results and Discussion

### 3.1. Characteristics of Chitosan and Alginate Solutions and SSD Stability

The versatility of 3D printing, allowing the production of complex structures [45], enables the exploration of this route as suggested by Boetker with respect to different excipients [46] and by Martinez et al. on solid dosage forms [47]. In this paper, the feasibility of preparing 3D printed polymer-based drug delivery systems was explored. 

As viscosity is a crucial parameter for the accurate deposition of 3D structures according to the design, the measurement of this parameter on the CH (18,500 cP ±30 cP at 60 rpm) and ALG (45,875 ± 28 cP at 60 rpm) solutions was initially carried out, proving to be adequate for our 3D printing system.

After the completion of the 3D printing procedure, SSD was loaded into the ALG hydrogels by exploiting a soaking technique [48], as reported in the Experimental section. ALG hydrogels were selected for the SSD loading as this polymer is widely used in active dressing for the treatment of chronic wounds [49]. The SSD was stable for up to 21 days in contact with the alginate hydrogel, showing a recovery of 102.0 ± 0.2%. The total amount of drug loaded by the ALG hydrogels was estimated by dissolving them and by measuring the drug absorbance at 305 nm. After 1 h, each hydrogel had reached its maximum loading ability (SSD content of 760 μg ± 65 μg; RSD 8.7%). No significant differences were found in the amount of loaded drug after 3 or more hours, suggesting that an equilibrium between the absorption and release of the drug by the hydrogel had been reached by 1 h of soaking. 

### 3.2. Physico-Chemical Characterization of Printed Hydrogels

#### 3.2.1. Evaluation of Water Content and Elasticity

As physico-chemical features of hydrogels could have a determining influence for controlling drug delivery, 3D printed hydrogels were deeply characterized. 

To characterize alginate and chitosan hydrogels, the water content was measured. The amount in ALG (93.18 ± 0.88%) and CH (92.76 ± 1.17%) hydrogels in their swollen status were not significantly different. This was considered a positive element for the association of these two hydrogels as drug delivery systems, as a water content imbalance between the different types of hydrogels could alter liquid movement within the system, with the consequent alteration of drug diffusion through it. Considering the volume of the ALG hydrogel (0.127 mL) and its water content (93.18% corresponding to 0.118 μL), the maximum amount of drug that could be loaded, by the complete substitution of water with drug solution, was estimated to be 885 μg. The efficiency of SSD loading at equilibrium was thus estimated to be 86 ± 3%. 

Elasticity is another important aspect when designing a drug delivery system, especially for application on soft tissues such as skin, both in terms of handling and adaptation to the area of application [50]. Young’s modulus of the ALG hydrogels reached 0.582 ± 0.019 Mpa, whereas that of CH resulted in 0.365 ± 0.015 Mpa. Student’s T-test confirmed that the difference in the elasticity values of the different polymeric hydrogels was statistically significant (*p* < 0.005). Traction tests put into evidence a statistically significant smaller elasticity of alginate hydrogels, as testified by the lower value of deformation with respect to the chitosan hydrogels, which corresponded to a weaker consistency of the gels. For this reason, taking advantage of chitosan’s mechanical resistance as well as of the possibility of loading SSD onto ALG, we combined 3D printed scaffolds formed from the two biomaterials in order to create tougher structures, easier to handle in view of a possible application as drug delivery systems, for example, for the treatment of infected wounds.

#### 3.2.2. SEM Analysis

In this work, ALG hydrogels with and without SSD were characterized morphologically by SEM (Figure 2) to evaluate the macrostructure of the 3D hydrogels. By comparing the reported images, it is possible to notice that the alginate hydrogel (Figure 2A) presented a well-defined 3D macrostructure with respect to the loaded one (Figure 2B), where the surface precipitation of the drug occurred after dehydration. In Figure 2C,D, the ALG hydrogel details are highlighted. Compared to the CH hydrogel (Figure 2E,F), both the surface and the cross-section presented a much more compact structure characterized by few and superficial not-interconnected pores.

CH hydrogels prepared by exposure to ammonia vapors showed a pore size distribution between 0.5 and 13 μm on the surface and between 2 and 38 μm in the cross-section (Figure 2E,F); on the other hand, the layer of chitosan obtained by freeze-casting showed a compact surface layer, as previously reported [51]. Morphological observation by SEM revealed that drug-loaded hydrogels were subjected to partial dissolution with a reduction in the amount of alginate. The mechanism of the gelation of alginate salts relies on the ionotropic cross-linking mediated by divalent cations, such as Ca^2+^ ions, but the stability of these gels in fluids is subject to the presence of other cations: monovalent cations, as silver ions associated with sulfadiazine during the loading procedure, can exchange with Ca^2+^ ions present in the alginic mesh, causing the progressive dissolution and loosening of the network [52]. The cross-linking reaction, macrostructure and inner porosity of the scaffolds and the mechanical resistance are strictly correlated as already described in the literature [53]. These results here obtained demonstrated that the features of the 3D constructs developed can fit with their main purpose to act as a biocompatible antimicrobic active dressing while ensuring suitable easy handling, space for oxygen exchange and controlled SSD release.

#### 3.2.3. ATR–FTIR Spectroscopy

To better understand the incorporation of the drug into the hydrogel, the ATR–FTIR spectra of the ALG hydrogel and the ALG-SSD-containing hydrogel were compared (Figure 3). 

The spectrum of the sodium alginate powder (Figure 3A) showed typical signals in the 4000–600 cm^−1^ wavelength interval. In particular, a wide peak at 3450 cm^−1^ typical of the stretching of the -OH group is present, as well as two peaks at 1618 and 1440 cm^−1^ for the -COO^−^ group and an acute peak at 1050 cm^−1^ for the C=O stretching. Analogously, the spectrum of the SSD powder (Figure 3B) revealed that characteristic absorbance peaks at 3390 and 3340 cm^−1^ referred to the symmetric and asymmetric stretching of the -NH_2_ groups, respectively. The group at 1655 cm^−1^ corresponds to the bending of the -NH_2_ group, whereas those at 1595 and 1500 cm^−1^ are related to the phenolic skeleton. The peaks observed at 1230 and 1130 cm^−1^ are characteristic of the asymmetric vibration of the SO_2_ group. Aromatic vibration appears at 1070 cm^−1^. 

The loading of the drug is demonstrated by the appearance, in the fingerprint zone, of peaks specific to SSD which are not present in the unloaded hydrogel. In particular, the peak at 1277 cm^−1^ that in the SSD powder spectrum appeared at 1230 and 1130 cm^−1^, which was characteristic of the asymmetric stretching vibration group, SO_2_. Such a shift towards lower wavenumbers might suggest the interaction of this SSD functional group with the polymeric matrix. Furthermore, in the SSD-containing hydrogel, two peaks appear compared to the alginate one, at 1139 and at 963 cm^−1^.

### 3.3. Release Tests

Four different hydrogel constructs (Figure 1) were designed and built in order to obtain different drug release profiles. Chitosan hydrogels were associated with drug-loaded alginate hydrogels with the purpose of conferring a stiff consistency to the structure as well as exploiting the useful chitosan characteristics in terms of biocompatibility and antimicrobial action. By performing release experiments, the four samples showed different release rates and overall amounts of released drugs. 

As is shown in Figure 4, the ALG hydrogel only showed the fastest release rate and the highest SSD amount released, with an initial burst release of the drug. 

During the first 60 min, 76 ± 14% of the loaded drug is released, reaching almost 100% after 3 h. When the alginate hydrogels were assembled with a thin chitosan film, the release of the drug was partially hampered, reaching a maximum of 56 ± 13% of the loaded drug after 5 h. Both the rate and cumulative amount of the released drug were lower with respect to the alginate gels.

The 5CA combination showed a maximum drug release of 72 ± 4% in 5 h. Such a difference with respect to alginate gel only can be attributable to the time needed by the solvent to diffuse through the chitosan layers and reach the alginate hydrogel. With respect to the AF construct, no lag time is observed, and no barrier to solvent absorption is exerted by the chitosan hydrogel, thanks to the macroporosity conferred by the 3D printing design. The SSD release occurring within a few minutes is analogous to that observed from the alginate-only constructs. This feature could be ascribed to the drug located on the surface of the alginate hydrogel, which is easily reached by SWF by capillary action. In an attempt to estimate the relation between the thickness of the chitosan hydrogel and the drug release, the 20CA construct was designed. Such a modification led to a consistent reduction in the amount of the released drug (48 ± 7% after 5 h) which was, on the whole, lower than the amount released by the AF construct. The release starts after an initial lag time, attributable to the time needed by SWF to wet the chitosan layers and reach the SDD-loaded alginate gel. After that, an almost linear drug kinetic is observed, characterized by a progressive reduction in the release rate. 

ALG scaffolds showed the fastest and highest release of SSD. Such a speed can be explained by the fact that the hydrogel tested is directly in contact with the membrane, allowing a fast absorption of SWF by capillary action through the whole thickness of the hydrogel, its full hydration and fast drug release by diffusion. The macropores formed by 3D deposition constitute a preferential way for SWF to reach the top of the hydrogel, leading to the fast diffusion of the drug that is located on the surface. The diffusion of SWF through hydrogel filaments could be responsible for the slower release of the drug trapped within the hydrogel observed in the other three constructs, as the drug had to follow a winding path before leaving the polymeric matrix. In the AF construct, a lag time was observed. This reduction in release rate can be ascribed to the fact that chitosan film is a compact structure acting as a barrier to drug diffusion, also delaying SWF capillary diffusion in alginate hydrogel.

All constructs with chitosan, both films and 3D printed hydrogels, lead to an overall reduction in the final amount of SSD released, which was inversely related to the thickness of the chitosan hydrogel. The overall amount of the released drug was statistically different with respect to the alginate scaffold as such, as well as among chitosan/alginate constructs. In all cases, a plateau was reached, and no further drug release was observed after 5 h (data not shown). This suggests that the barrier effect created by the presence of chitosan could be attributable not only to physical segregation, strongly prevailing in chitosan film, but also to molecular interactions occurring between SSD and the polymer that prevents drug release and is proportional to the amount of chitosan encountered by the drug along its diffusion path. The water content analysis reported above led us to exclude that differences in drug delivery could be ascribed to a different hydration state of hydrogels.

From a therapeutic point of view, the amounts of SSD released are all above the reported MIC (minimal inhibitory concentration) of SSD (ranging from 50 to 100 μg/mL, depending on the bacteria [54]), suggesting that hydrogels could be effective, if applied on wounds, in controlling bacterial proliferation. 

### 3.4. Cytotoxicity Tests

Cytotoxicity tests were conducted in order to evaluate the effect of SSD on cell vitality in relation to the different release behavior of the constructs. In general, the results demonstrated that the vitality of the cells in contact with the constructs loaded with the drug is much lower. Compared to the control (no scaffold), resorufin fluorescence in the presence of scaffold A showed a reduction to 60.8 ± 1.5%, 48 ± 17% for 5CA, 60 ± 8% for AF and 59 ± 4% for 20CA constructs (*p* < 0.05) after 24 h of incubation (Figure 5). The data observed for all the constructs indicate that within 24 h, the release of SSD causes suffering to the cells. This observation is in agreement with the data reported in the literature, reporting the cytotoxic effect of SSD on in vitro cell cultures [55]. In vitro studies have also shown that SSD cytotoxicity can be reduced by controlling the administration of the active agent [56]: based on the release experiments, it was expected that the major cytotoxicity would have been determined by A scaffolds, as these constructs were characterized by the major quantity of SSD released and in the shortest time, as previously described. From data obtained after 3 h of incubation, the construct causing more cytotoxicity is the 5CA (Figure 5). Moreover, the cytotoxicity of the four different constructs was very similar. To explain these results, it is necessary to take into consideration that the conversion of resazurin to resorufin, used for the vitality tests, occurs by an oxidation reaction, which is also one of the main mechanisms of SSD degradation. The observed results are thus probably determined by a combined effect related not only to the intrinsic SSD cytotoxicity but also to its degradation. 

### 3.5. Kirby–Bauer Antimicrobial Activity Assay

Finally, antimicrobial activity assays were carried out to test the potential efficacy of the 3D printed hydrogel prototypes against *S. aureus* and *P. aeruginosa*. 

As expected, the antimicrobial assay demonstrated that the controls (ALG scaffolds not containing SSD) did not show any antimicrobial activity. The SSD–ALG scaffold, which is the core of all developed constructs, demonstrated the capability of inhibiting the growth of both resistant bacteria species. After 18–24 h of incubation, the inhibition ring was measured, reaching a 10 mm diameter for *S. aureus* (Figure 6A) and a 9 mm diameter for *P. aeruginosa* (Figure 6B), probably due to the drug diffusion through the agar layer and highlighting the high sensibility of the bacteria to SSD. This supports the hypothesis of using these hydrogels in combination with chitosan scaffolds for the treatment of chronically infected wounds, which would benefit from a slower and more sustained release. 

## 4. Conclusions

In conclusion, in this study, a 3D printing technique was used to produce CH/ALG-based hydrogels with four different architectures to be tested as drug delivery systems for the controlled release of SSD. Morphological studies demonstrated the suitability and precision of the 3D printing technology combined with ALG and CH inks to create well-controlled hydrogel structures with mechanical resistance suitable for the easy handling of the scaffolds. ALG and CH mechanical resistance could be further improved by adding additional co-polymers (i.e., nanocellulose) to the initial inks, opening the way to future investigations. The overall release of the drug is above the MIC, and the efficacy of the 3D constructs with the drug embedded in the composite scaffolds in terms of antimicrobial activity was demonstrated against Gram+ and Gram- bacteria. The use of different constructs resulted in significant changes in terms of the rate and amount of SSD released, offering a key for the tailoring of drug administration over time as a function of the wound-healing stage. In general, considering the in vitro toxicity of the drug on fibroblast cells, it is desirable to have a slower rate of SSD release. Therefore, the profile obtained with SSD-loaded alginate gels coupled with 20 layers of chitosan suggests the potential sustainable application of this inexpensive dressing for further in vivo tests. As infected wound healing involves complex biological processes as a function of the wound environment, the efficacy of these scaffolds of the antimicrobic/regenerative effects should require deeper in vivo tests.

The results shown here provided the basis for the development and improvement of sustainable personalized active medications that are dose- and kinetic-demanding for patient-tailored therapy. The advantages of 3D printing technology were demonstrated and could be deeply investigated and exploited for the manufacturing of SSD-loaded hydrogels with controlled release properties by modifying scaffold geometries, mechanical resistance, polymeric ink, post-printing processing steps, dressing toxicity, etc., with the capability to afford the overall healthcare needs required by our society. 

## Figures and Tables

**Figure 1 micromachines-14-00137-f001:**
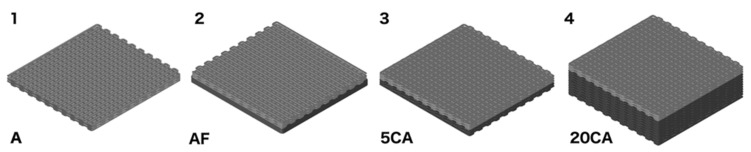
Schematic representation of 3D constructs. Light grey represents SSD-loaded alginate hydrogels, and dark grey represents chitosan hydrogels: (**1**) 3D printed 5-layer alginate loaded with SSD alone (A); (**2**) 3D printed 5-layer alginate + chitosan film (AF) (**3**) 3D printed 5-layer chitosan hydrogel + 3D printed 5-layer alginate (5CA); (**4**) 3D printed 20-layer chitosan hydrogel + 3D printed 5-layer alginate (20CA).

**Figure 2 micromachines-14-00137-f002:**
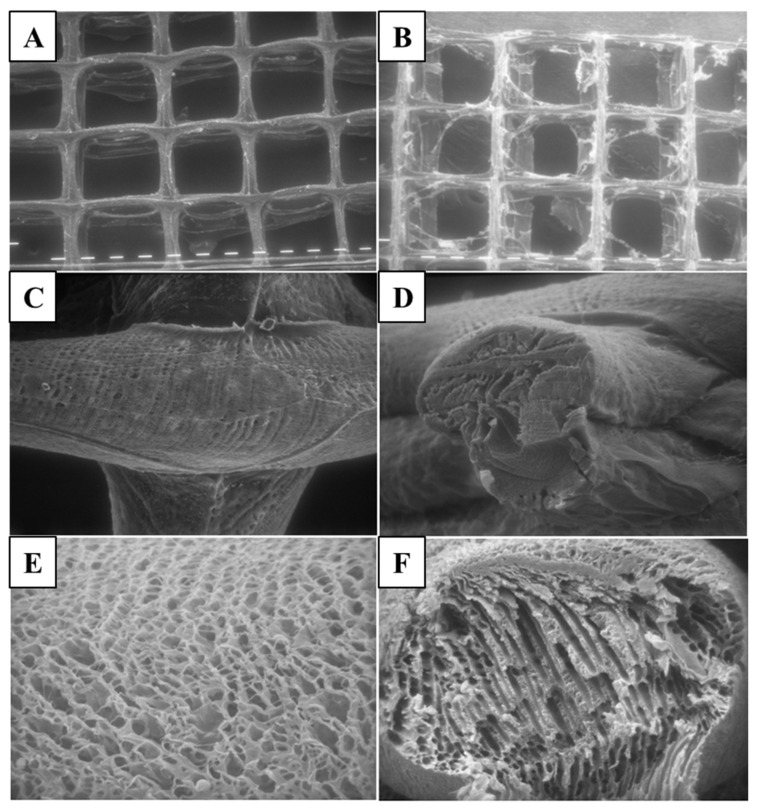
SEM pictures of 3D hydrogels. Calcium alginate hydrogel (**A**) without SSD and (**B**) containing SSD (magnification 40X); calcium alginate hydrogel surface (**C**) and cross-section (**D**) (magnification 320X); (**E**) chitosan hydrogel gelled in ammonia vapor surface (magnification 1250X) and (**F**) cross section (magnification 640X).

**Figure 3 micromachines-14-00137-f003:**
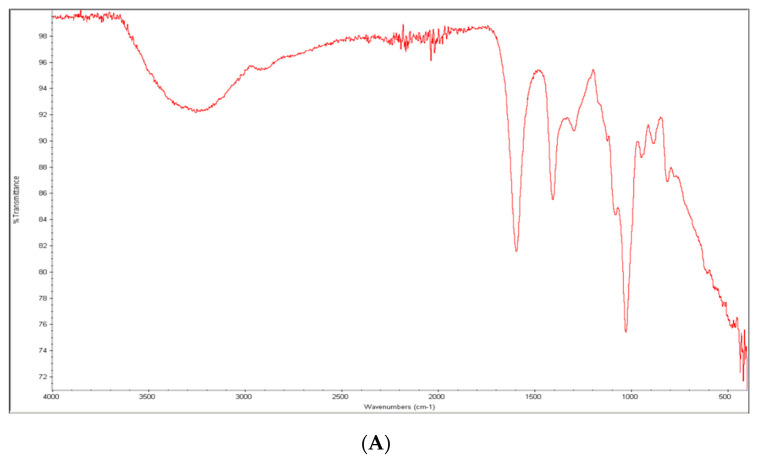
ATR–FTIR spectra. (**A**) sodium alginate; (**B**) silver sulfadiazine powders; (**C**) comparison between the ATR–FTIR spectra of alginate hydrogel (in red) and the alginate SSD-containing hydrogel (in green).

**Figure 4 micromachines-14-00137-f004:**
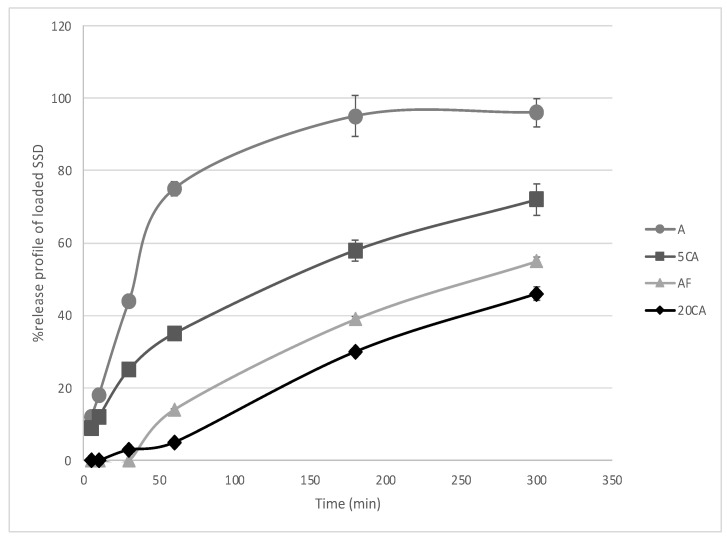
Percent release of loaded SSD from different hydrogel combinations: A (circle), AF (triangle); 5CA (square), 20CA (rhombus). Release tests (n = 3) were performed at 37 °C in simulated wound fluid [42] composed of CaCl_2_ anhydrous 0.02 M; NaCl 0.4 M, tris methylamine 0.08 M (final pH 7.5).

**Figure 5 micromachines-14-00137-f005:**
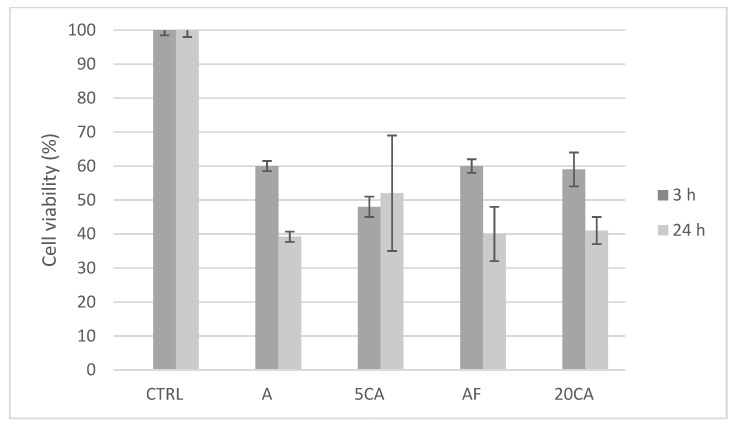
Effect of the four 3D printed hydrogel constructs on fibroblast cell viability. Cell viability was determined by resazurin assay at 3 h (dark gray lines) and 24 h (light gray lines) on the negative control (CTRL). The results are reported as means ± standard deviation of three independent measurements.

**Figure 6 micromachines-14-00137-f006:**
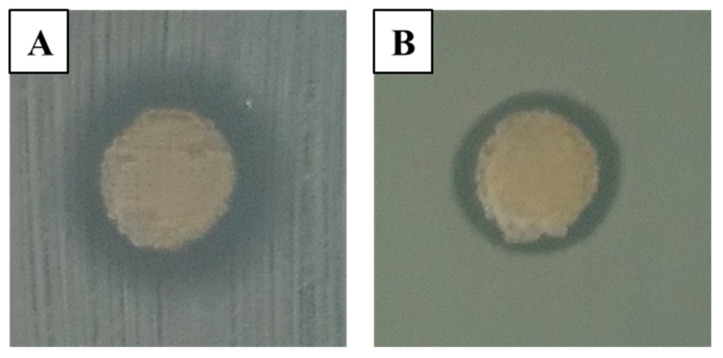
Kirby–Bauer antimicrobial activity assay against multidrug-resistant strains of *S. aureus* (**A**) and *P. aeruginosa* (**B**).

## Data Availability

The authors declare that the data generated and analyzed during this study are included in this published article. In addition, datasets generated and/or analyzed during the current study are available from the corresponding author on reasonable request.

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
