# Peer review of "3D Printed Chitosan/Alginate Hydrogels for the Controlled Release of Silver Sulfadiazine in Wound Healing Applications: Design, Characterization and Antimicrobial Activity"

_micromachines, 2023, doi:10.3390/mi14010137_

Round 1
Reviewer 1 Report
In this study, the authors created 3D printed chitosan/alginate-based scaffolds containing silver sulfadiazine for wound healing applications. The manuscript is well-designed; however, major revisions are required. The manuscript should be revised in response to the comments.
1. The application of the prepared scaffolds (e.g., wound healing) should be mentioned in the title of the paper.
2. The "Abstract" is poorly written. The research findings should be mentioned. The results of cytotoxicity tests. The "Abstract" also lacks a clear conclusion.
3. The authors should provide some background on wound healing and skin tissue engineering in the "Introduction" section.
4. The physicochemical and biological properties of chitosan (for the reference: https://onlinelibrary.wiley.com/doi/full/10.1002/jbm.b.35039) and alginate (for the reference: https://www.mdpi.com/1999-4923/10/2/42) that make them suitable biopolymers for wound healing applications should be discussed in the "Introduction" section.
5. The version of all software used in this study should be specified.
6. On page 3, lines 110-112, how did the authors ensure that the gelling agent was completely removed?
7. A relevant reference is required for the formula used in subsection "2.3. Water content and elasticity."
8. On page 4, lines 148 and 149, the gold coater device information should be included.
9. In the "Materials and Methods" section, the authors should include a separate subsection explaining the statistical analysis used in this study.
10. The mechanical properties findings are poorly discussed. The authors should discuss the findings by comparing the mechanical properties of the prepared scaffolds to those of other chitosan/alginate-based scaffolds from other studies.
11. The significance of porosity should be discussed further in subsection "3.2. Physico-chemical characterization of printed hydrogels." The porosity of a scaffold is critical for providing adequate spaces for cell accommodation, proliferation, migration, and differentiation. Porous scaffolds also aid in the oxygenation and nutrition of the injured skin. (For the reference: https://jnanobiotechnology.biomedcentral.com/articles/10.1186/s12951-020-00755-7)
12. How many replicates were used for the release tests?
13. The silver sulfadiazine-containing scaffolds have exhibited significant cytotoxicity. How can the authors justify the scaffold's suitability for wound healing applications in terms of biological properties?
14. It would be helpful if the authors included a chart showing the cytotoxicity results.
15. A clear concluding statement is required at the end of the "Conclusions" section.
16. Most of the references are not up-to-date. The most recently published articles should be used to discuss the obtained results.
Other minor revisions:
- The words "in vitro," "Staphylococcus aureus," and "Pseudomonas aeruginosa" should be written in italics in the "Abstract."
- "Wound healing" should be considered as a keyword.
- Part "c" of figure 3 should be defined in the caption.
Author Response
The file with the reply to the Referee's comment was added.

Reviewer 2 Report
The authors have submitted an interesting article " 3D printed chitosan/alginate hydrogels for the controlled release of silver sulfadiazine: design, characterization and antimicrobial activity" which deals with the preparation and characterization of different 3d printed chitosan (CH)/alginate (ALG) based platforms intended as antimicrobial active dressings for wound healing. The manuscript is well structured and reads well overall, although it will need some revisions. I suggest this article be published after a serious major revision.
*** General comments:
ü The abstract is clear and concise and comprises all cornerstones including a brief/general introduction to the topic, a non-technical summary of the major findings, and their implications.
ü The introduction is compelling, clear, and concise. The introduction part covers a proper description of the challenge/gap and a strong background in the field associated with a fair literature review, however, it can be improved further.
ü The various sections of the body of the text are clear and concise overall.
ü The experimental design is logical, however, there are still some comments to be covered and some concerns to be addressed.
ü The conclusions are logical.
*** Suggested revisions:
1- First of all, I strongly recommend the authors provide a simple, high-quality, and informative) “Graphical abstract” which can present the whole concept of your study at a glance. I would like to recommend authors design a “Graphical Abstract” for this study to better show the entire story in a simple and informative manner. In this regard, you can illustrate a simple sketch of the big picture and add elements like Macroscopic photos of platforms, printing process, SEM images, etc. (totally up to you). You can easily use the “Biorender” website.
2- Please carefully revise the manuscript to remove grammatical errors and vague sentences. Some of the sentences are unnecessary, making it difficult and boring for the readers to follow them. Moreover, please keep consistent with labeling figures (Use only capital or small letters for different sections of all figures), and please label the figure section at the top left of each one for all figures (check figures 3 and 5)
Figure 4, Y-axis “% Release of loaded SSD”.
Please double-check the whole manuscript and revise all.
3- One of the best ways to highlight the outcomes of a study is to tabulate a comparative master table to compare the findings of this study with recently published data from other research groups. I suggest the authors add a such table to this manuscript.
4- Some of the references are too old (e.g., 2004, 2005, etc.). A myriad of research bodies has been published in recent years and you can find similar concepts and cite them in your paper. Moreover, in the introduction part to better present the fundamentals of this field, please read and add valuable information from the following key paper as well:
“Chitosan-based inks for 3D printing and bioprinting - https://doi.org/10.1039/D1GC01799C “
5- I recommend the authors present the cytotoxicity results in the form of a bar graph which is a better way to show the data.
6- The infected area is always associated with slightly acidic pH due to the activity of immune cells. Based on this fact, I was wondering what was the pH of media in Release test. Please add all details ( pH and temperature (37°C)) to the (Figure 4) caption.
7- The conclusion section is short and insufficient. Please develop it based on the obtained data.
Author Response
The reply to Referee's comment were added as file.

Round 2
Reviewer 1 Report
The corrections and additions introduced by the authors improved the structure and quality of the manuscript. I have no further suggestions.
Reviewer 2 Report
The manuscript is well-amended and it is ready for publication.